# Vaccination Coverage in Adult Patients with Inflammatory Bowel Disease: Impact of a Tailored Vaccination Pathway Including COVID-19 and Herpes Zoster in a University Hospital Vaccination Center

**DOI:** 10.3390/vaccines13090961

**Published:** 2025-09-11

**Authors:** Roberto Venuto, Caterina Elisabetta Rizzo, Daniela Lo Giudice, Walter Fries, Concetta Ceccio, Francesco Fedele, Raffaele Squeri, Cristina Genovese

**Affiliations:** 1Department of Chemical, Biological, Pharmaceutical and Environmental Sciences, University of Messina, 98166 Messina, Italy; 2Department of Prevention, Local Health Authority of Messina, 98123 Messina, Italy; caterina.rizzo93@gmail.com; 3Department of Biomedical and Dental Sciences and Morphofunctional Imaging, University of Messina, 98124 Messina, Italy; dlogiudice@unime.it (D.L.G.); ceccioc@unime.it (C.C.); squeri@unime.it (R.S.); crigenovese@unime.it (C.G.); 4Department of Clinical and Experimental Medicine, University of Messina, 98124 Messina, Italy; 5Hospital Hygiene Operative Unit, University Hospital of Messina “G. Martino”, 98124 Messina, Italy; f.fedele1965@libero.it

**Keywords:** hospital vaccination center, vaccination coverage, inflammatory bowel diseases

## Abstract

**Background/Objectives:** Patients with inflammatory bowel disease (IBD) are at increased risk of severe infections, particularly when undergoing immunosuppressive therapy. Vaccination is a key preventive strategy, but coverage in this group is often suboptimal. This study evaluated vaccination coverage among IBD patients at diagnosis/referral and after admission to a structured hospital-based vaccination pathway. **Methods**: We conducted an observational study (February 2022–February 2025) at the Vaccination Center (VC) of the University Hospital “G. Martino” in Messina, Italy. Adult IBD patients referred by gastroenterologists were assessed for vaccination status using hospital and regional registries, and personalized schedules were developed based on Italian National Vaccine Prevention Plan guidelines. Descriptive statistics were applied to assess baseline and post-intervention vaccination coverage. **Results**: Of 154 participants (mean age 64 years; 51.9% male), 55.4% were on immunosuppressive therapy. Baseline coverage was heterogeneous: influenza, 6.5%; PCV13, 25.5%; PPV23, 26.6%; herpes zoster, 62.3%; and COVID-19 primary cycle, 79.6%. After enrollment, substantial improvements were observed: influenza, 89.2%; PCV13, 74.5%; PPV23, 67.0%; herpes zoster, 75.4%; and COVID-19 primary cycle, 96.8%. Coverage for catch-up vaccines also improved (e.g., HBV went from 1.9% to 44.2%). However, uptake of COVID-19 booster doses during the study period remained low (15.6%). No significant differences emerged by sex or treatment subgroup. **Conclusions**: A structured, collaborative care pathway between gastroenterologists and public health specialists significantly improved vaccination coverage among IBD patients. Despite gains, gaps persist in COVID-19 booster uptake and catch-up vaccinations. Integration of vaccination services into routine IBD management is essential to enhance protection in this high-risk population.

## 1. Introduction

Vaccination remains one of the most effective public health interventions for preventing infectious diseases, protecting both individuals and the broader community through herd immunity. Preventive vaccination strategies have particular relevance for patients with inflammatory bowel diseases (IBDs), with proven benefits in reducing comorbidities, hospitalization rates, and mortality [1]. The management of IBD also imposes a substantial economic burden on healthcare systems due to frequent and prolonged hospitalizations, particularly when complications occur [2].

IBD refers to chronic inflammatory conditions of the gastrointestinal tract, representing a major public health challenge [3,4,5,6]. Global data show that IBD prevalence is higher in regions with a high socio-demographic index (SDI), partly due to better healthcare access and diagnostic capacity [7,8]. In Italy, despite the absence of a centralized national IBD registry, estimates suggest that more than 200,000 people are affected. In Sicily, 2023 data from hospital discharge records and exemption codes indicated a prevalence approximately 40% higher than the national average—about 300 cases per 100,000 people [9,10].

This apparent paradox—higher prevalence in Sicily, a region not typically classified as high SDI—may reflect local diagnostic practices (e.g., reliance on administrative databases), environmental or genetic factors, or differences in healthcare-seeking behavior and disease awareness. These findings emphasize the need to consider both methodological and socioeconomic aspects when interpreting epidemiological data.

Treatment for IBD often involves anti-inflammatory and immunosuppressive drugs, with biologic therapies such as monoclonal antibodies (e.g., adalimumab) increasingly used [11]. While effective for disease control, these therapies substantially increase infection risk [12,13]. Patients with IBD have up to a tenfold higher risk of developing severe infections compared to the general population [14].

Patients receiving TNF-alpha inhibitors are particularly susceptible to infections including Legionella pneumophila, Epstein–Barr virus (EBV), hepatitis B virus (HBV), cytomegalovirus (CMV), and varicella zoster virus (VZV). These can manifest as pneumonia, post-herpetic neuralgia, or hemophagocytic lymphohistiocytosis [15]. Influenza and pneumococcal infections also occur more frequently and can be more severe in IBD patients [14,16]. Advanced age further compounds these risks: in IBD patients >65 years old, opportunistic and severe infections, hospitalizations, and mortality occur at higher rates compared to younger patients [17].

Given these concerns, early assessment of vaccination status and tailored vaccination schedules are crucial for IBD patients, ideally before initiating immunosuppressive therapy [18]. According to Italian national guidelines and international best practices, indicated vaccines should be administered at least four weeks prior to starting immunosuppression. If therapy has already begun, vaccination may still proceed following an individualized risk–benefit assessment [17]. This study aimed to evaluate vaccination coverage among IBD patients at the following points:(a)At the time of diagnosis/referral.(b)After admission to a hospital-based vaccination center that implemented a structured clinical pathway jointly developed by gastroenterologists and public health specialists.

The hypothesis is that the implementation of a structured hospital-based vaccination pathway would significantly improve coverage across multiple recommended and catch-up vaccines in adult IBD patients, with particular emphasis on those most relevant to immunosuppressive therapy, such as influenza, pneumococcal, herpes zoster, and COVID-19 vaccines.

## 2. Materials and Methods

### 2.1. Study Design and Setting

We conducted an observational study over 3 years (February 2022–February 2025) at the vaccination center (VC) of the University Hospital “G. Martino” of Messina (UHM), Italy. This period was chosen to capture vaccine coverage data relevant to COVID-19 booster doses and herpes zoster vaccination. The mean follow-up time of enrolled patients was 14.2 months (SD = 5.7), and the last patient was included in February 2024, allowing for at least 12 months of follow-up for the majority of the cohort.

The study was based on a collaboration protocol between the Hospital Hygiene Unit and the Gastroenterology/IBD Unit at UHM, which manages approximately 250 IBD patients. The protocol, established in 2018, was designed to improve vaccine uptake in this high-risk population.

### 2.2. Care Pathway

The intervention consisted of the following: (i) after IBD diagnosis, referral by gastroenterologists to the VC; (ii) review of vaccination history and eligibility by public health specialists—physicians with a four-year postgraduate specialty in Public Health—based on patient records, the Italian National Vaccine Prevention Plan (PNPV), and regional vaccination registries; (iii) development of a personalized vaccination plan, considering immunosuppressive treatment schedules; (iv) patient counseling including infection-risk education, printed reminders, and coordination with treating physicians; (v) encouragement strategies such as vaccine co-administration to minimize visits; and (vi) follow-up reviews at least annually, or earlier if clinically indicated.

At our center, all vaccination plans were reviewed by senior specialists in Public Health in collaboration with gastroenterologists.

### 2.3. Inclusion Criteria

Eligible participants were

≥18 years old;Diagnosed with IBD;Capable of providing informed consent;Consenting to vaccination at the VC.

Of all IBD patients referred to the VC, 62.4% provided consent and were enrolled. A significant proportion declined or did not consent.

The study followed the ethical principles outlined in the 1996 version of the Declaration of Helsinki and Good Clinical Practice guidelines. All patients provided written informed consent before enrollment. The study was notified to the Sicilian Regional Ethics Committee and recorded in an ad hoc logbook with the number 511400, with approval granted on 4 February 2025.

### 2.4. Vaccination Data Sources

Baseline vaccination coverage was assessed using multiple sources:Hospital vaccination records (paper and electronic);Regional vaccination registry (ONIT, Metropolitan City of Messina);“Vax-Center” and “OnVac” platforms for SARS-CoV-2 and other vaccines.

Importantly, only documented vaccine doses were counted as vaccinated. Serological evidence of immunity (e.g., varicella IgG) was not considered equivalent to vaccination for this study.

### 2.5. Vaccine Eligibility

We used the described guidelines and the summaries of product characteristics of the recommended vaccines for the study population to draw up an ad hoc schedule for every patient. As reported in Table 1, the main indications for every vaccine until to date are extracted by the Italian National Vaccine Prevention Plan [18], the annual Circulars about the prevention and control of influenza [19], and the Circulars about recommendations for anti-COVID-19 vaccination campaign [20].

In order to promote adherence to vaccinations by reducing the number of visits to the VC and to optimize immunization times, a strategy that was used, if there were no contraindications, as indicated by the PNPV 2023–2025 [18], was that of co-administering several vaccines in the same vaccination session (for example, flu and pneumococcal vaccinations, flu and COVID-19 vaccinations).

For immunocompromised patients, accelerated schedules were applied (e.g., recombinant zoster vaccine second dose after 1–2 months instead of 2–6).

In addition to the vaccines recommended for IBD patients, we offered to this population the possibility to catch up on vaccinations addressed to the general population or other categories if they have missed a vaccination or did not receive the recommended number of doses (Table 2).

In Italy, all vaccines recommended for IBD patients by the PNPV (influenza, pneumococcal, herpes zoster, varicella, and COVID-19) are fully covered by the National Health Service. Catch-up vaccinations (HBV, HPV, DTPa, MMRV) are also covered when indicated by age or clinical condition, though HPV and MMRV in adults may depend on regional prioritization.

Not all patients were eligible for each vaccine. Eligibility was determined by age, clinical status, and national guidelines. For example:Pneumococcal vaccines: both PCV13 and PPV23 were administered according to age and immunosuppression status.HPV vaccine: primarily recommended for younger adults; however, catch-up doses were offered according to PNPV guidance. Very few older patients (e.g., >50 years) were vaccinated for HPV, and only in cases where individual risk justified off-label use.Herpes zoster vaccine: recommended for adults ≥50 years or immunosuppressed individuals.

### 2.6. Statistical Analysis

Categorical variables were expressed as frequencies and percentages, while continuous variables were summarized as means and standard deviations (SD). Comparisons of vaccination coverage between subgroups were performed using Chi-square tests or Fisher’s exact tests when expected cell counts were <5. A two-sided *p*-value < 0.05 was considered statistically significant. All analyses were performed using R software version 4.4.0.

## 3. Results

Of the patients admitted to the Vaccination Center (VC), 62.4% (n = 154) met the inclusion criteria and agreed to participate in the study.

The gender distribution was nearly balanced: 51.9% were male (n = 80) and 48.1% female (n = 74). The mean age of participants was 64 years (SD = 17.37), with the most represented age group being 45–64 years (45.4%). Patients ranged in age from 18 to 90 years.

In terms of treatment, 55.4% (n = 85) were undergoing immunosuppressive therapy at the time of enrollment, while 5.3% (n = 8) were being treated with erlotinib. The remaining patients were either on 5-aminosalicylates or not receiving any pharmacological treatment at the time of the study. All patients completed the study follow-up.

Of eligible patients, 37.6% declined participation. For ethical reasons, no demographic or clinical information was collected from these individuals.

Table 3 provides a breakdown of the study population by age group and treatment category.

### 3.1. Vaccination Coverage at the Time of Admission to the VC

Vaccination coverage at baseline was heterogeneous. Higher coverage was observed for some vaccines—such as herpes zoster (62.3%) and the COVID-19 primary cycle (79.6%)—while others showed substantially lower rates. For example, influenza vaccination was 6.5%, and pneumococcal vaccines had limited coverage (13-valent pneumococcal conjugate vaccine, PCV13: 25.5%; 23-valent pneumococcal polysaccharide vaccine, PPV23: 26.6%). Coverage for catch-up vaccinations (HBV, HPV, DTPa, MMRV) was also very low at the time of enrollment.

### 3.2. Vaccination Coverage After Admission to the VC

By the end of the study, vaccination coverage had increased substantially across nearly all categories. The following coverage rates were achieved: influenza, 89.2% (n = 137); PCV13, 74.5% (n = 114); PPV23, 67.0% (n = 103); and Herpes Zoster (HZV), 75.4% (n = 116).

Regarding COVID-19, 96.8% of participants had completed the primary vaccination series. However, only 15.6% of the total sample had received an additional booster dose during the study period. The observed decrease in booster dose coverage compared to baseline indicates that the reported figure (15.6%) only includes booster doses administered during the study period and does not account for all booster doses participants had received previously before enrollment.

Figure 1 illustrates vaccination coverage before and after admission to the Vaccination Center, highlighting improvements for almost each vaccine. These results are consistent with the detailed values reported in Table 4.

### 3.3. Comparison of Vaccination Coverage Before and After Admission to the VC

Table 4 summarize changes in vaccination coverage for both recommended and catch-up vaccines before and after enrollment in the VC pathway.

Baseline and post-intervention vaccination coverage was compared by sex (Table 5) and by treatment group (Table 6) to explore subgroup differences. Statistical testing showed no significant differences in coverage improvements across any vaccine type.

## 4. Discussion

This study aimed to assess vaccination coverage in patients with inflammatory bowel disease (IBD) admitted to a university hospital vaccination center (VC). The findings show a heterogeneous pattern of vaccine uptake at baseline, followed by a substantial overall improvement in coverage after admission to the VC. However, some critical gaps remain—particularly with respect to COVID-19 booster doses.

A key strength of this study was the relatively high participation rate (62.4%) among eligible patients, with a balanced gender distribution and a broad age range. The average age was 64 years, with nearly half of the cohort (45.4%) falling within the 45–64-year age group. This age profile is clinically relevant, as older adults with IBD are at increased risk of vaccine-preventable infections, hospitalization, and adverse outcomes, particularly when immunocompromised.

The study also drew on multiple data sources (hospital records, regional registries, and digital platforms), ensuring a robust assessment of vaccine coverage.

At baseline, vaccination coverage varied widely. Coverage for the COVID-19 primary vaccination cycle and herpes zoster vaccine was relatively high, likely reflecting strong public health messaging during the COVID-19 pandemic and increased awareness surrounding herpes zoster risk in older adults. In contrast, coverage for more routine vaccinations—such as influenza and pneumococcal vaccines—was considerably lower. These findings align with existing literature that highlights persistent vaccination gaps among immunocompromised and chronically ill populations [21]. Factors potentially contributing to lower baseline coverage include limited patient education, lack of clear recommendations from treating physicians, and underestimation of infection risks by patients [22,23].

Not all patients were candidates for every vaccine: for example, HPV vaccination was mainly relevant for younger adults, although in a few cases older individuals were offered catch-up doses on the basis of clinical judgment.

Following the implementation of the VC-based intervention, vaccination coverage for nearly all recommended and catch-up vaccines improved significantly. Influenza vaccine uptake rose from 6.5% to 89.2%, and pneumococcal vaccine coverage (PCV13 and PPV23) increased to 74.5% and 67%, respectively. Herpes zoster vaccine coverage also improved from 62.3% to 75.4%. These results underscore the effectiveness of targeted, multidisciplinary vaccination programs in improving immunization rates in high-risk groups such as IBD patients. The increase in herpes zoster vaccination, in particular, may also reflect the growing availability and acceptance of the recombinant zoster vaccine (RZV), which is considered safer and more effective than earlier formulations [24,25,26,27,28].

Despite these positive trends, the low uptake of COVID-19 booster doses remains a significant concern. While the primary COVID-19 vaccination series was completed by 96.8% of participants, only 15.6% received an additional booster dose during the study period. This apparent decline in booster coverage compared to pre-admission reflects that our data specifically capture booster doses administered during the study period, not lifetime booster history. Nonetheless, the finding highlights a potential gap in ongoing protection for IBD patients—especially those on immunosuppressive therapy—given their heightened vulnerability to severe COVID-19 and emerging variants [29,30].

The low uptake of booster doses may be explained by several factors, including inadequate patient counseling, evolving national recommendations, vaccine hesitancy, and barriers to access [31,32]. These results indicate a need for more proactive follow-up strategies, clearer communication, and stronger integration between gastroenterology and vaccination services to ensure that eligible patients receive timely booster doses.

The limited improvement observed for DTPa and MMRV catch-up vaccinations contrasts with the marked gains achieved for IBD-recommended vaccines. Several factors may explain this discrepancy. First, DTPa and MMRV are generally considered childhood or adolescent vaccines, and many adult patients perceived them as less relevant to their immediate health needs. Second, clinical prioritization may have favored vaccines more directly associated with immunosuppressive therapy (e.g., pneumococcal, influenza, herpes zoster, and COVID-19), resulting in lower emphasis on catch-up vaccines during counseling. Finally, logistical aspects such as multi-dose schedules (particularly for MMRV) and lack of routine reminders may have contributed to reduced adherence. These findings suggest that specific communication strategies are needed to emphasize the importance of catch-up immunization in adulthood, even among high-risk patients.

Interestingly, the data did not reveal significant differences in vaccination uptake between patients receiving immunosuppressive therapy and those not on such treatment. This may suggest that the centralized vaccination model used in this study was effective in promoting consistent vaccine uptake across treatment subgroups. However, more granular subgroup analyses—particularly by age, disease severity, or type of immunosuppressant—could provide deeper insights into how treatment status interacts with vaccine adherence.

### Limitations of the Study

Several limitations of this study should be acknowledged. Almost 40% of eligible patients declined to participate, and due to ethical restrictions no information was collected from this group; in addition, the reasons for non-consent were not systematically collected. This lack of information is relevant, as excluding this group may have led to an overestimation of vaccination uptake among the broader IBD population.

The assessment of baseline vaccination status relied primarily on documented records from hospital and regional registries. Vaccinations received outside the hospital system may not have been captured, and serological evidence of immunity was not considered. Together, these factors may have underestimated true baseline protection.

Although our pathway addressed a broad panel of vaccines, the most clinically emphasized components were influenza, pneumococcal, herpes zoster, and COVID-19, reflecting their higher priority in immunosuppressed IBD patients. Results for other vaccines such as DTPa and MMRV should be interpreted with caution given the smaller number of eligible adults and lower overall uptake.

Finally, the single-center observational design imposes restrictions on the strength of causal inference. Without a control group, improvements in vaccination uptake cannot be conclusively attributed to the VC pathway alone. Moreover, the relatively small and geographically localized sample limits generalizability and reduces the ability to detect subgroup differences.

## 5. Conclusions

Vaccination is a cornerstone of infection prevention in patients with IBD, particularly those undergoing immunosuppressive therapy. Our study demonstrated that integrating a structured, hospital-based vaccination pathway into routine IBD care significantly increased coverage across recommended vaccines, including influenza, pneumococcal, herpes zoster, and COVID-19. These findings confirm our hypothesis that a multidisciplinary, patient-centered approach enhances vaccine uptake in this high-risk population, in line with previous reports [33,34].

Despite these improvements, coverage for catch-up vaccines such as DTPa and MMRV remained suboptimal, reflecting lower perceived relevance, logistical barriers, and competing clinical priorities [35,36,37]. Addressing these gaps requires targeted communication and tailored strategies to emphasize the importance of adult catch-up immunization [38,39,40,41]. Overall, our results highlight that embedding vaccination services into specialized care pathways can substantially improve protection against vaccine-preventable diseases in adults with IBD, while also contributing to antimicrobial resistance reduction [42,43].

## Figures and Tables

**Figure 1 vaccines-13-00961-f001:**
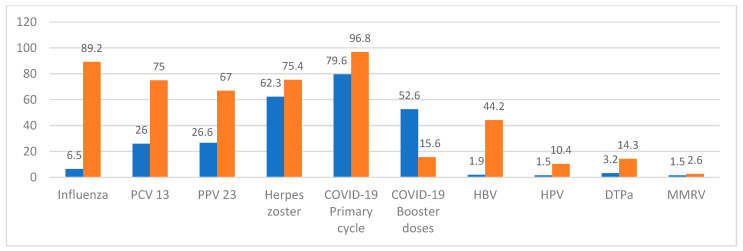
Vaccination coverage in percentages at the baseline (in blue) and at end of the study (in orange) (n = 154).

**Table 1 vaccines-13-00961-t001:** Recommended vaccinations for IBD patients.

Vaccine	Document	Target Population	Schedule	Notes
Influenza	Annual Circulars of the Italian Ministry of Health “Prevention and control of influenza”	“subjects aged 6 months to 65 years with pathologies that increase the risk of complications from influenza: h) chronic inflammatory diseases and intestinal malabsorption syndromes”; “people aged ≥ 65 years”	1 dose annually	Also recommended for family members, cohabitants and caregivers of people with serious frailties
Pneumococcal	PNPV	“Conditions requiring long-term immunosuppressive treatment”	“number of doses as per technical sheet depending on age and pathology or condition”	
Herpes Zoster	PNPV	“Subjects with congenital/acquired immunodeficiency or intended for immunosuppressive therapy”	Two doses (0, 2–6 months). In subjects who are or may become immunodeficient or immunosuppressed due to disease or therapy and who would benefit from an accelerated schedule, the second dose of RZV may be administered 1 to 2 months after the initial dose”	“provided that the adjuvanted recombinant vaccine (RZV) is used”
Varicella	PNPV	“Subjects intended for immunosuppressive therapy”	“two doses at least 4 weeks apart”	
COVID-19	Circulars of the Italian Ministry of Health about recommendations for anti-COVID-19 vaccination campaign	“People aged 6 months to 59 years inclusive, with high fragility, as they suffer from pathologies or conditions that increase the risk of severe COVID-19:—Chronic inflammatory diseases and intestinal malabsorption syndromes” “people aged ≥ 65 years”	2 doses (primary cycle) followed by booster doses	Also recommended for family members, cohabitants and caregivers of people with serious frailties

**Table 2 vaccines-13-00961-t002:** Catch-up vaccinations for IBD patients.

Vaccine	Document	Target Population	Schedule
HBV	PNPV	“Vaccination of all previously unvaccinated adults”	Three doses (0, 1, 6 months)
HPV	PNPV	Starting from 11 years	>15 years: “3-dose vaccination cycle at 0, 2, 6 months starting at age 15”
DTPa	PNPV	“people aged ≥18 years”	Every 10 years
MMRV	PNPV	“For all subjects who have not been vaccinated with two doses”	Generally, 2 doses administered at least 28 days apart from each other

**Table 3 vaccines-13-00961-t003:** Characteristics of the study population at baseline (n = 154).

Characteristic	n (%)
**Sex**	
Male	80 (51.9%)
Female	74 (48.1%)
**Age group**	
18–29	10 (6.5%)
30–44	24 (15.6%)
45–64	70 (45.4%)
≥65	50 (32.5%)
**Treatment**	
Immunosuppressive therapy	85 (55.4%)
Erlotinib	8 (5.3%)
Other/no treatment	61 (39.6%)

**Table 4 vaccines-13-00961-t004:** (**A**) Changes in coverage for recommended vaccinations in IBD patients; (**B**) Changes in coverage for catch-up vaccinations.

Vaccine	Pre-Admittance (%)	Post-Admittance (%)	Δ Coverage
(**A**)			
Influenza *	6.5% (10)	89.2% (137)	+82.7%
PCV 13	25.5% (39)	74.5% (114)	+49%
PPV 23	26.6% (41)	67% (103)	+40.4%
Herpes Zoster	62.3% (96)	75.4% (116)	+13.1%
COVID-19 Primary cycle	79.6% (123)	96.8% (149)	+17.2%
COVID-19 Booster doses **	52.6% (81)	15.6% (24)	−37%
(**B**)			
HBV	1.9% (3)	44.2% (68)	+42.3%
HPV	1.5% (2)	10.4% (16)	+8.9%
DTPa	3.2% (5)	14.3% (22)	+11.1%
MMRV	1.5% (2)	2.6% (4)	+1.1%

* influenza coverage refers to the flu season preceding the first VC access. ** data refer only to booster doses administered during the study period, not total lifetime booster coverage.

**Table 5 vaccines-13-00961-t005:** Vaccination coverage by gender at the baseline and the end of the study.

Vaccine	Pre-Admittance		Post-Admittance		*p*-Value
	M (n = 80)	F (n = 74)	M (n = 80)	F (n = 74)	
Influenza	7.5% (6)	5.4% (4)	88.8% (71)	89.2% (66)	0.88
PCV 13	26.3% (21)	24.3% (18)	72.5% (58)	75.7% (56)	0.64
PPV 23	27.5% (22)	25.7% (19)	68.8% (55)	64,9% (48)	0.57
HZV	63.8% (51)	60.8% (45)	72,5% (58)	78,4% (58)	0.47
COVID-19 Primary cycle	80.0% (64)	79.1% (59)	96.3% (77)	97.3% (72)	0.73
COVID-19 Booster doses	53.8% (43)	51.4% (38)	15.0% (12)	16.2% (12)	0.84
HBV	2.5% (2)	1.4% (1)	45% (36)	43,2% (32)	0.78
HPV	2.5% (2)	0% (0)	8.8% (7)	12.2% (9)	0.66
DTPa	3.8% (3)	2.7% (2)	15% (12)	13.5% (10)	0.82
MMRV	1.3% (1)	1.4% (1)	2.5% (2)	2.7% (2)	0.95

**Table 6 vaccines-13-00961-t006:** Vaccination coverage stratified by treatment group at the baseline and the end of the study.

Vaccine	Immunosuppressive Therapy (n = 85)		Erlotinib (n = 8)		Other/No Treatment (n = 61)		*p*-Value
	Pre	Post	Pre	Post	Pre	Post	
Influenza	5.9% (5)	88.2% (75)	6.3% (1)	87.5% (7)	6.6% (4)	90.2% (55)	0.93
PCV13	24.7% (21)	72.9% (62)	25.0% (2)	75.0% (6)	26.2% (16)	75.4% (46)	0.94
PPV23	27.1% (23)	64.7% (55)	25.0% (2)	87.5% (7)	26.2% (16)	67.2% (41)	0.42
Herpes Zoster	62.4% (53)	74.1% (63)	75.0% (6)	75.0% (6)	60.7% (37)	77.0% (47)	0.92
COVID-19 Primary cycle	80.0% (68)	96.5% (82)	87.5% (7)	100% (8)	78.7% (48)	96.7% (59)	0.86
COVID-19 Booster doses	54.1% (46)	15.3% (13)	50.0% (4)	12.5% (1)	50.8% (31)	16.4% (10)	0.95
HBV	2.4% (2)	44.7% (38)	0.0% (0)	37.5% (3)	1.6% (1)	42.6% (26)	0.76
HPV	2.4% (2)	8.2% (7)	0.0% (0)	12.5% (1)	0.0% (0)	13.1% (8)	0.62
DTPa	3.5% (3)	14.1% (12)	0.0% (0)	12.5% (1)	3.3% (2)	14.8% (9)	0.98
MMRV	1.2% (1)	2.4% (2)	0.0% (0)	0.0% (0)	1.6% (1)	3.3% (2)	0.84

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
