# Peer review of "Vaccination Coverage in Adult Patients with Inflammatory Bowel Disease: Impact of a Tailored Vaccination Pathway Including COVID-19 and Herpes Zoster in a University Hospital Vaccination Center"

_vaccines, 2025, doi:10.3390/vaccines13090961_

Round 1

Reviewer 1 Report

Comments and Suggestions for Authors

This is a useful report on the results of a tailored vaccination pathway offered to adult patients with inflammatory bowel disease at a hospital-based vaccination unit in Sicily.

Some modifications/clarifications are needed in order to improve the clarity and the usefulness of this experience for the potential readers, as follows:

  1. Please, include in the title that the study was restricted to adult patients.
  2. Please, inform the mean follow-up time of the included patients and the date of the last patient included in the study. This is relevant to fully appreciate the actual period of patients´ exposure to the intervention.
  3. Please, exclude the comment on the risk of selection bias in the methods section. The same comment was already stated in the discussion section.
  4. Figures 1 and 2 should be just one, including the baseline and post intervention vaccination coverage in order to facilitate the comparison of the improvement achieved for each vaccine. Also, the data showed in those figures as redundant with the data showed in Table 5a.
  5. It is difficult to appreciate the usefulness of the results showed in table 4 because of the lacking of data coverage at baseline stratified by sex. I suggest the inclusion of the baseline coverage data in table 4. Also, if any statistical comparison were made, the significance values should be demonstrated. The meaningful comparison should be between the proportions of coverage improvement observed by sex for each vaccine as the authors calculated in table 5. Indeed, the exploration of subgroups stratified by sex should be better demonstrated after Table 5a that contains the main study findings.
  6. Please, discuss the potential reasons for the lower coverage improvement for the DTPa and MMRV vaccines. It is hard to understand the successful improvement for the specific IBD recommended vaccines in contrast to the very poor results for DTPa and MMRV vaccines.

Author Response

Thanks for your comments. Below a point-to-point response.

1) Please, include in the title that the study was restricted to adult patients.

We modified the title in "Vaccination Coverage in Adult Patients With Inflammatory Bowel Disease: Impact of a Tailored Vaccination Pathway Including COVID-19 and Herpes Zoster in a University Hospital Vaccination Center"

2) Please, inform the mean follow-up time of the included patients and the date of the last patient included in the study. This is relevant to fully appreciate the actual period of patients' exposure to the intervention.

In section 2.1. we added "The mean follow-up time of enrolled patients was 14.2 months (SD = 5.7), and the last patient was included in February 2024, allowing for at least 12 months of follow-up for the majority of the cohort."

3) Please, exclude the comment on the risk of selection bias in the methods section. The same comment was already stated in the discussion section.

We removed it from Methods, left in Discussion/Limitations.

4) Figures 1 and 2 should be just one, including the baseline and post intervention vaccination coverage in order to facilitate the comparison of the improvement achieved for each vaccine. Also, the data showed in those figures as redundant with the data showed in Table 5a.

We replaced Figures 1 and 2 with a single Figure 1 showing baseline and post-intervention coverage side by side. We removed redundancy by adjusting the text: "Figure 1 illustrates vaccination coverage before and after admission to the Vaccination Center, highlighting improvements for almost each vaccine. These results are consistent with the detailed values reported in Table 4 (former Table 5)."

5) It is difficult to appreciate the usefulness of the results showed in table 4 because of the lacking of data coverage at baseline stratified by sex. I suggest the inclusion of the baseline coverage data in table 4. Also, if any statistical comparison were made, the significance values should be demonstrated. The meaningful comparison should be between the proportions of coverage improvement observed by sex for each vaccine as the authors calculated in table 5. Indeed, the exploration of subgroups stratified by sex should be better demonstrated after Table 5a that contains the main study findings.

We expanded Table 5 (former Table 4) to include baseline coverage by sex, adding statistical significance column. We relocated this table. We also wrote "Baseline and post-intervention vaccination coverage was compared by sex to explore subgroup differences (Table 5). Statistical testing showed no significant differences in coverage improvements between male and female participants across any vaccine type."

6) Please, discuss the potential reasons for the lower coverage improvement for the DTPa and MMRV vaccines. It is hard to understand the successful improvement for the specific IBD recommended vaccines in contrast to the very poor results for DTPa and MMRV vaccines.

We added at the end of Discussion, just before Limitations: "The limited improvement observed for DTPa and MMRV catch-up vaccinations contrasts with the marked gains achieved for IBD-recommended vaccines. Several factors may explain this discrepancy. First, DTPa and MMRV are generally considered childhood or adolescent vaccines, and many adult patients perceived them as less relevant to their immediate health needs. Second, clinical prioritization may have favored vaccines more directly associated with immunosuppressive therapy (e.g., pneumococcal, influenza, herpes zoster, and COVID-19), resulting in lower emphasis on catch-up vaccines during counseling. Finally, logistical aspects such as multi-dose schedules (particularly for MMRV) and lack of routine reminders may have contributed to reduced adherence. These findings suggest that specific communication strategies are needed to emphasize the importance of catch-up immunization in adulthood, even among high-risk patients."

Reviewer 2 Report

Comments and Suggestions for Authors

The manuscript submitted by Roberto Venuto and co-authors to Vaccines is devoted to the effect of two vaccines on the IBD in patients on immunosuppressive therapy

Some issues should be resolved prior to the article might be recommended for publication

1) Please decipher all abbreviations used in Fig. 1 and Fig. 2 captions, and remove .00%, since they are not required

2) Change the title and add the information related to the COVID-19 and herper zoster vaccination

3) Describe the hypothesis in Introduction section and state in the Conclusions, whether it was proven or refuted

4) Reduce the size of the Conclusion to 1-2 paragraphs, since in its current form its content repeats the Discussion section

5) Add information to section 4.1 that only two vaccines were considered

Author Response

Thanks for your suggestions. Point-by-point response below.

1) Please decipher all abbreviations used in Fig. 1 and Fig. 2 captions, and remove .00%, since they are not required.

We explained the abbreviations used in Fig. 1 in the text above and we removed unnecessary decimals.

2) Change the title and add the information related to the COVID-19 and herper zoster vaccination

Since the study covers a full panel of vaccines (influenza, pneumococcal, HBV, HPV, etc.), we changed the title in "Vaccination Coverage in Adult Patients With Inflammatory Bowel Disease: Impact of a Tailored Vaccination Pathway Including COVID-19 and Herpes Zoster in a University Hospital Vaccination Center".

3) Describe the hypothesis in Introduction section and state in the Conclusions, whether it was proven or refuted.

In the Introduction we added “The hypothesis is that the implementation of a structured hospital-based vaccination pathway would significantly improve coverage across multiple recommended and catch-up vaccines in adult IBD patients, with particular emphasis on those most relevant to immunosuppressive therapy, such as influenza, pneumococcal, herpes zoster, and COVID-19 vaccines."

In the Conclusions: “Our study demonstrated that integrating a structured, hospital-based vaccination pathway into routine IBD care significantly increased coverage across recommended vaccines, including influenza, pneumococcal, herpes zoster, and COVID-19. These findings confirm our hypothesis that a multidisciplinary, patient-centered approach enhances vaccine uptake in this high-risk population, in line with previous reports."

4) Reduce the size of the Conclusion to 1-2 paragraphs, since in its current form its content repeats the Discussion section.

We reduced the Conclusions section as follows "Vaccination is a cornerstone of infection prevention in patients with IBD, particularly those undergoing immunosuppressive therapy. Our study demonstrated that integrating a structured, hospital-based vaccination pathway into routine IBD care significantly increased coverage across recommended vaccines, including influenza, pneumococcal, herpes zoster, and COVID-19. These findings confirm our hypothesis that a multidisciplinary, patient-centered approach enhances vaccine uptake in this high-risk population, in line with previous reports.

Despite these improvements, coverage for catch-up vaccines such as DTPa and MMRV remained suboptimal, reflecting lower perceived relevance, logistical barriers, and competing clinical priorities. Addressing these gaps requires targeted communication and tailored strategies to emphasize the importance of adult catch-up immunization. Overall, our results highlight that embedding vaccination services into specialized care pathways can substantially improve protection against vaccine-preventable diseases in adults with IBD, while also contributing to antimicrobial resistance reduction."

5) Add information to section 4.1 that only two vaccines were considered.

In section 4.1. we added "Although our pathway addressed a broad panel of vaccines, the most clinically emphasized components were influenza, pneumococcal, herpes zoster, and COVID-19, reflecting their higher priority in immunosuppressed IBD patients. Results for other vaccines such as DTPa and MMRV should be interpreted with caution given the smaller number of eligible adults and lower overall uptake."

Reviewer 3 Report

Comments and Suggestions for Authors

Well-conceived, well- written and relevant study, of interest to public health, vaccinologists and gastro-enterologists. Strengths and limitations appropriately discussed.

Author Response

We sincerely thank Reviewer 3 for the positive feedback on our work. No specific modifications were requested, but we carefully reviewed the manuscript once more to ensure clarity and accuracy throughout.

Reviewer 4 Report

Comments and Suggestions for Authors

As a general comment, the manuscript VACCINATION COVERAGE IN PATIENTS WITH INFLAM-MATORY BOWEL DISEASE: IMPACT OF A TAILORED VAC-CINATION PATHWAY IN A UNIVERSITY HOSPITAL VAC-CINATION CENTER provides interesting data on the impact of vaccination program implemented for IBD patients and provides some valuable information which should be considered by public health professionals and decision makers.

In my opinion, the content and quality of the paper is good, however, it should improve by considering the following issues:

From public health perspective it would be worth to know what vaccinations for IBD are covered in Italy by the national health insurance plan/s and what are recommended but not covered – this may explain the vaccination coverage. My feeling is that all vaccinations presented are covered but it is not clearly said.

It would be worth knowing whether public health specialists had medical background, in some countries majority of these specialists are not physicians. What kind of training should they undergo to be eligible to develop the vaccination plan. Was there any senior specialist who reviewed this?

As there was a remarkable proportion of IBD patients who refused participation it should be provided clinical and demographic characteristic of these patients with a comparisons and comments on the possibility of systematic sampling bias in the study.

There is no information on whether any calculations of sample size were done, what were assumptions for sample size. If there were no sample size considerations I suggest providing post hoc analysis of study power.

I suggest providing the Institutional Review Board / Ethics Committee acceptance number

As a baseline characteristic I suggest providing a table showing vaccination coverage by the treatment received -> this may show the differences and may show patient’s clinical futures associated with low vaccination coverage. The same tabulation I suggest for the end of study data.

It would be worth knowing what was the main follow-up time for a patient

Provide, please, information on what type of vaccination schemes (if any) were used, especially for immuno-compromised patients

Among most important, I believe, authors should describe details of the VC-based intervention, point-by-point steps of intervention, strategies for control, methods of encouragements or stimuli, counseling or communication strategies used, frequency of check-ups.

Under ‘Limitations’ … as I understand the purpose of the study was to assess the impact of tailored vaccination pathway, if so, it is hard to say, that the presence of patients who were contraindications for a particular vaccine / or do not fulfill the criteria to get it is a limitation of the study (like it is described for HPV). If authors want to rise that issue I suggest moving it to general discussion.

Under ‘Conclusions’ there is a continuation of discussion. I suggest moving majority of that part into ‘Discussion’ and leave there only a conclusion statement.

Reviewer

Author Response

From public health perspective it would be worth to know what vaccinations for IBD are covered in Italy by the national health insurance plan/s and what are recommended but not covered – this may explain the vaccination coverage. My feeling is that all vaccinations presented are covered but it is not clearly said.

We clarified which vaccines are covered by the Italian NHS. In 2.5 section we added "In Italy, all vaccines recommended for IBD patients by the PNPV (influenza, pneumococcal, herpes zoster, varicella, and COVID-19) are fully covered by the National Health Service. Catch-up vaccinations (HBV, HPV, DTPa, MMRV) are also covered when indicated by age or clinical condition, though HPV and MMRV in adults may depend on regional prioritization."

It would be worth knowing whether public health specialists had medical background, in some countries majority of these specialists are not physicians. What kind of training should they undergo to be eligible to develop the vaccination plan. Was there any senior specialist who reviewed this?

We added clarification. in 2.2. section "... public health specialists - physicians with a four-year postgraduate specialty in Public Health - based on patient records, the Italian National Vaccine Prevention Plan (PNPV), and regional vaccination registries; ..." and "At our center, all vaccination plans were reviewed by senior specialists in Public Health in collaboration with gastroenterologists."

As there was a remarkable proportion of IBD patients who refused participation it should be provided clinical and demographic characteristic of these patients with a comparisons and comments on the possibility of systematic sampling bias in the study.

Unfortunately, due to ethical constraints, we could not collect or analyze detailed demographic or clinical data from patients who did not provide consent. We acknowledge that this prevents direct comparison between participants and non-participants. In section 3 we added "Of eligible patients, 37.6% declined participation. For ethical reasons, no demographic or clinical information was collected from these individuals."

There is no information on whether any calculations of sample size were done, what assumptions were for sample size. If there were no sample size considerations I suggest providing post hoc analysis of study power.

Given the observational design, no formal a priori sample size calculation or post hoc power analysis was performed.

I suggest providing the Institutional Review Board / Ethics Committee acceptance number

In 2.3 section: "The study followed the ethical principles outlined in the 1996 version of the Declaration of Helsinki and Good Clinical Practice guidelines. All patients provided written informed consent before enrollment. The study was notified to the Sicilian Regional Ethics Committee and recorded in an ad hoc logbook with the number 511400, with approval granted on February 4, 2025."

As a baseline characteristic I suggest providing a table showing vaccination coverage by the treatment received -> this may show the differences and may show patient’s clinical futures associated with low vaccination coverage. The same tabulation I suggest for the end of study data.

We added a new table (Table 6. Vaccination coverage stratified by treatment group at the baseline and the end of the study.)

It would be worth knowing what was the main follow-up time for a patient

In 2.1 section: "The mean follow-up time of enrolled patients was 14.2 months (SD = 5.7), and the last patient was included in February 2024, allowing for at least 12 months of follow-up for the majority of the cohort."

Provide, please, information on what type of vaccination schemes (if any) were used, especially for immuno-compromised patients

In 2.5 section we wrote "For immunocompromised patients, accelerated schedules were applied (e.g., recombinant zoster vaccine second dose after 1–2 months instead of 2–6)."

Among most important, I believe, authors should describe details of the VC-based intervention, point-by-point steps of intervention, strategies for control, methods of encouragements or stimuli, counseling or communication strategies used, frequency of check-ups.

We expanded description in 2.2. section "The intervention consisted of: (i) after IBD diagnosis, referral by gastroenterologists to the VC; (ii) review of vaccination history and eligibility by public health specialists - physicians with a four-year postgraduate specialty in Public Health - based on patient records, the Italian National Vaccine Prevention Plan (PNPV), and regional vaccination registries; (iii) development of a personalized vaccination plan, considering immunosuppressive treatment schedules; (iv) patient counseling including infection-risk education, printed reminders, and coordination with treating physicians; (v) encouragement strategies such as vaccine co-administration to minimize visits; and (vi) follow-up reviews at least annually, or earlier if clinically indicated. At our center, all vaccination plans were reviewed by senior specialists in Public Health in collaboration with gastroenterologists."

Under ‘Limitations’ … as I understand the purpose of the study was to assess the impact of tailored vaccination pathway, if so, it is hard to say, that the presence of patients who were contraindications for a particular vaccine / or do not fulfill the criteria to get it is a limitation of the study (like it is described for HPV). If authors want to rise that issue I suggest moving it to general discussion.

We moved and reframed in general discussion "Not all patients were candidates for every vaccine: for example, HPV vaccination was mainly relevant for younger adults, although in a few cases older individuals were offered catch-up doses on the basis of clinical judgment."

Under ‘Conclusions’ there is a continuation of discussion. I suggest moving majority of that part into ‘Discussion’ and leave there only a conclusion statement.

We reduced Conclusions as follows: "Vaccination is a cornerstone of infection prevention in patients with IBD, particularly those undergoing immunosuppressive therapy. Our study demonstrated that integrating a structured, hospital-based vaccination pathway into routine IBD care significantly increased coverage across recommended vaccines, including influenza, pneumococcal, herpes zoster, and COVID-19. These findings confirm our hypothesis that a multidisciplinary, patient-centered approach enhances vaccine uptake in this high-risk population, in line with previous reports [33-34]. Despite these improvements, coverage for catch-up vaccines such as DTPa and MMRV remained suboptimal, reflecting lower perceived relevance, logistical barriers, and competing clinical priorities [35-37]. Addressing these gaps requires targeted communication and tailored strategies to emphasize the importance of adult catch-up immunization [38–41]. Overall, our results highlight that embedding vaccination services into specialized care pathways can substantially improve protection against vaccine-preventable diseases in adults with IBD, while also contributing to antimicrobial resistance reduction [42–43]."

Round 2

Reviewer 1 Report

Comments and Suggestions for Authors

Authors have done all the requested modifications/clarifications.

Author Response

Thanks for your comments.

Reviewer 2 Report

Comments and Suggestions for Authors

The manuscipt has been improved and might be recommended for publication

Author Response

Thanks for your comments.

Reviewer 4 Report

Comments and Suggestions for Authors

Dear Authors,

Thank you for all the corrections and improvements made in the submitted manuscript. In my opinion the quality of the content has been nicely improved.  Provide, please, the name of statistical test using for the analyses presented in Table 6.

Overall, in my opinion, the manuscript is now well prepared to be published.

Reviewer

Author Response

We thank you for your positive evaluation of our revised manuscript and for your final helpful comment. Regarding Table 6, the statistical analysis was performed using the Chi-square test (or Fisher’s exact test when expected cell counts were <5) to compare vaccination coverage between subgroups. This information has now been specified in the Statistical analysis subsection of the Methods.

"Categorical variables were expressed as frequencies and percentages, while continuous variables were summarized as means and standard deviations (SD). Comparisons of vaccination coverage between subgroups were performed using Chi-square tests or Fisher’s exact tests when expected cell counts were <5. A two-sided p-value <0.05 was considered statistically significant. All analyses were performed using R software."